# KNOWLEDGE ACCUMULATION IN CONTINUALLY LEARNED REPRESENTATIONS AND THE ISSUE OF FEATURE FORGETTING

## ABSTRACT

While it is established that neural networks suffer from catastrophic forgetting "at the output level", it is debated whether this is also the case at the level of representations. Some studies ascribe a certain level of innate robustness to representations, that they only forget minimally and no critical information, while others claim that representations are also severely affected by forgetting. To settle this debate, we first discuss how this apparent disagreement might stem from the coexistence of two phenomena that affect the quality of continually learned representations: knowledge accumulation and feature forgetting. We then show that, even though it is true that feature forgetting can be small in absolute terms, newly learned information is forgotten just as catastrophically at the level of representations as it is at the output level. Next we show that this feature forgetting is problematic as it substantially slows down knowledge accumulation. We further show that representations that are continually learned through both supervised and self-supervised learning suffer from feature forgetting. Finally, we study how feature forgetting and knowledge accumulation are affected by different types of continual learning methods.

## 1 INTRODUCTION

Machine learning models typically learn from static datasets and once they are trained and deployed, they are usually not updated anymore. Sometimes models make mistakes. Sometimes they do not work in a domain that was not trained. Sometimes they do not recognize certain classes or corner cases. Whatever the cause, sometimes it is necessary to update a model. The default choice in industry is to retrain a model from the beginning with new and old data to overcome malfunctions (Komolafe, 2023). Retraining a full model is costly and time-consuming, especially in deep learning. The goal of continual learning is to enable models to train continually, to learn from new data when they become available. This has proven to be a hard challenge (De Lange et al., 2022; van de Ven et al., 2022), as deep learning models that are continually trained exhibit catastrophic forgetting (McCloskey & Cohen, 1989; Ratcliff, 1990). Without precautionary measures, new data are learned at the expense of forgetting earlier acquired knowledge.

The data to train machine learning models rarely come in a format that is adapted to the problem we intend to solve. Taking the example of visual data, it is near impossible to infer higher-level properties directly from an image's raw pixel values. Hence, a first step is usually to transform them into a *representation* that makes solving the problem at hand an easier job. Often deep neural networks are used for this (Bengio et al., 2013). These networks learn semantically meaningful representations indirectly while optimizing their parameters to learn an input-output mapping. Sometimes the representation itself is the goal, yet often it is a final layer, or head, that uses the learned representation to assign an output (*e.g.* a class label) to an input. Even though they are commonly trained in unison, it can be useful to think of the representation and the head as two separate entities, working together.

In continual learning, there are at least two good reasons to care about representations. First, a strong representation makes it easier to learn new information. When a model already has a good representation of new data, it will require less changes to fully adapt to new data. This makes the re-use of existing features more likely, in turn lowering the risk of overwriting them, which

increases the risk of forgetting (Cha et al., 2021). Second, progressively accumulating knowledge from individual tasks into one representation may be a goal on its own. True continual learning should be able to use new information to its benefit and build a stronger representation over time, which can finally be used to solve a variety of tasks (Bengio et al., 2013).

It is with these motivations that recent work has been studying how representations are learned in continual learning, and how they forget. Among the researched topics are the effect of the depth of a layer on forgetting and learning (Ramasesh et al., 2021; Kim & Han, 2023) and the apparent robustness of representations to forgetting (Davari et al., 2022; Zhang et al., 2022). These works offer interesting insights, but they do not agree and open questions remain. Davari et al. (2022) write: "*[...] in many commonly studied cases of catastrophic forgetting, the representations under naive finetuning approaches, undergo minimal forgetting, without losing critical task information.*" and Zhang et al. (2022) similarly write: "*there seems to be no catastrophic forgetting in terms of representations*". Yet in similar experimental setups Kim & Han (2023) identify "*severe catastrophic forgetting*".

Another open question concerns whether feature forgetting, if it happens, hinders the learning of good representations. When studying the representation of continual learners using a downstream task (*i.e.* one that was not trained), Zhang et al. (2022) conclude that "*learning representations and catastrophic forgetting are largely separate issues*" and "*common techniques for mitigating catastrophic forgetting [...] have little effect on improving [representations]*". Similar conclusions are drawn by Cha et al. (2022). This suggests that only task-specific features might be forgotten. If this were true, feature forgetting would only be a problem if you care about the performance on the trained tasks, but not if you care about learning a good general representation. Yet, in the same papers, it is shown that learning many tasks together results in a better general representation than learning those same tasks one after the other, which seems to contradict that only task-specific features are forgotten.

Given these unresolved issues in the literature about forgetting and learning in continual representations, we aim to answer two questions:

**Question 1**: *Do continually trained representations forget catastrophically?*

With extensive experiments, we show that also at the level of representations, when training on new tasks, that what was learned during a past task is abruptly and greatly forgotten, or as it is called in literature: catastrophically. This leads us to the follow-up question:

**Question 2**: *Does it matter that these representations are forgotten?*

To test the impact of feature forgetting on the quality of the continually learned representation for downstream tasks, we compare the representation of a continually trained model against a representation that is ensembled from copies of the model after it is trained on each task. This ensemble baseline has a substantially better general representation than the continually trained model, showing that preventing feature forgetting is not only important for the performance on tasks that a model was trained on, but also for optimal knowledge accumulation.

Most experiments in this paper study the learning and forgetting of representations in supervised learning, but we show that our answers to the above two questions also hold for representations learned with self-supervised learning. We conclude the paper by evaluating examples of important families of continual learning methods and report how they influence the learning and forgetting of representations.

In summary our contributions include[1]:

- We show that continually learned respresentations do forget catastrophically (Section 3).
- We show that such forgetting in the representation negatively affects knowledge accumulation (Section 4).
- We compare feature forgetting and knowledge accumulation in different types of continual learning methods (Section 5).
- We show that self-supervised and contrastive learners suffer from feature forgetting as well (Section 5).

---

[1]Code will be made public upon acceptance.

## 2 PROBLEM STATEMENT AND EVALUATION

We follow the common definition of a continual learning setting by assuming a stream $\mathcal{T} = \{\mathcal{T}_1, \mathcal{T}_2, ..., \mathcal{T}_T\}$ of $T$ disjoint tasks $\mathcal{T}_i$. Each task consists of training data $X_i$ and targets $Y_i$, as well as respective test data $\hat{X}_i$, $\hat{Y}_i$. During training on each task the model has free access to the training data of that task, but not to the data of other tasks. Exception are replay memories, which can store small subsets of data from past tasks. On this stream of tasks we continually train a model $f_\theta$, with the goal to learn a model that works well for all tasks. Because, in this work we are particularly interested in how models continually learn and adapt a representation from sequence $\mathcal{T}$, we split the model into a shared backbone that produces the representation with parameters $\theta_B$, and a head with task-specific parameters $\theta_H = \{\theta_{h_1}, ..., \theta_{h_T}\}$ that utilizes the representation to solve the tasks.

Our main focus is on classification tasks. To measure continual learning performance in the standard way, we define $\text{ACC}_{i,j}$ as the test accuracy (the percentage of correctly classified test samples) on task $\mathcal{T}_j$ obtained by the model after training on task $\mathcal{T}_i$. We refer to this as *output* accuracy. Additionally, and central to this work, we explicitly evaluate the quality of the continually learned representations. Inspired by representation learning literature (Bengio et al., 2013; Chen et al., 2020; Zeiler & Fergus, 2014), we define the metric *linear probe accuracy*, denoted $\text{LP}_{i,j}$. After finishing training on task $\mathcal{T}_i$, a new set of parameters $\theta_{h_j}$ for the head's parameters of task $T_j$ are first trained with all training data in $\mathcal{T}_j$ while the backbone parameters $\theta_B$ are frozen. $\text{LP}_{i,j}$ is the test accuracy of the resulting model on task $\mathcal{T}_j$. The metric $\text{LP}_{i,j}$ thus measures the true suitability of the model's representation with respect to task $\mathcal{T}_j$ after training up to task $\mathcal{T}_i$. Lastly, when evaluating on a downstream task, *i.e.* one that was not part of training, this is indicated by $\text{LP}_{i,d}$. There are other ways to evaluate representations, *e.g.* using $k$-Nearest Neighbours, which we briefly review in the Supplemental. This did not lead to different conclusions, hence reporting in the main paper uses linear probes, as is common in the related literature.

In the main paper, the reported results are on Split MiniImageNet, a 20 task (5 classes each) split of MiniImageNet (Vinyals et al., 2016). The first 19 tasks are used as the training sequence, while the remaining task is never seen during training and used exclusively as a downstream task, to evaluate the quality of the representation. To reduce the influence of the inherent difficulty of a particular task, we use five different task sequences and report mean and standard errors on all results. The sequences are randomly generated but consistent across experiments. In the Supplemental material, we replicate all our results using a 10 task sequence of CIFAR-100 (Krizhevsky et al., 2009). For more details, see Supplemental.

## 3 REPRESENTATIONS FORGET CATASTROPHICALLY

To answer whether representations forget catastrophically, we need to comprehend what *"catastrophically"* refers to. For this, we turn to the two works that are often credited for discovering the phenomenon of catastrophic forgetting. McCloskey & Cohen (1989) note: "*[t]raining on a new set of items may drastically disrupt performance on previously learned items*", and Ratcliff (1990) describes this as: "*well-learned information is forgotten rapidly as new information is learned*". To be considered 'catastrophic', forgetting should thus be both 'drastic' and 'rapid'. We further note that, implicitly, both of the above descriptions consider the information that was learned during a task as what can be forgotten, respectively: "*previously learned items*" and "*well-learned information*". This is perhaps most clear in the definition of Robins (1993), inspired by the two earlier works: "*[i]f after its original training is finished a network is exposed to the learning of new information, then the originally learned information will typically be greatly disrupted or lost*". Summarized, catastrophic forgetting refers to the drastic and rapid forgetting of previously learned knowledge.

In recent papers, following Lopez-Paz & Ranzato (2017), forgetting is often calculated as the difference in performance immediately after training task $i$ and after training on a final task $n$. With $r_{i,j}$ the performance of task $j$ after training on task $i$, this becomes: $r_{i,i} - r_{n,i}$. They do not quantify how much forgetting would be considered catastrophic, and neither will we, but we can compare forgetting in the representation to that at the output layer, which is often identified as catastrophic. Note that $r$ can refer to any performance measure, so both output accuracy (ACC) and linear probing accuracy (LP) are a valid option. Forgetting, defined as such, only depends on the difference in performance relative to immediately after a task was trained, regardless of how much information

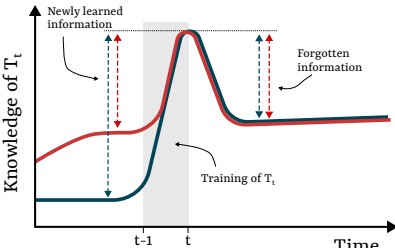

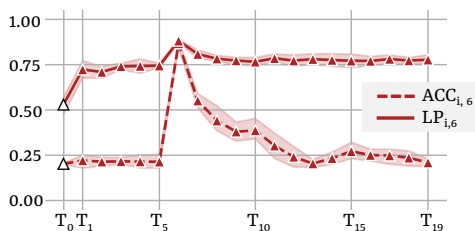

Figure 1: An illustration of why it matters to account for the learned information when calculating forgetting. Without one could conclude that both examples forget an equal amount. While the red example actually forgets everything it had learned, and the blue one only about 50%.

Figure 2: Linear probe and output accuracy of $T_6$ during the entire Mini-ImageNet sequence. $T_0$ indicates at model initialization, so before any training took place. (Mean $\pm$ standard error)

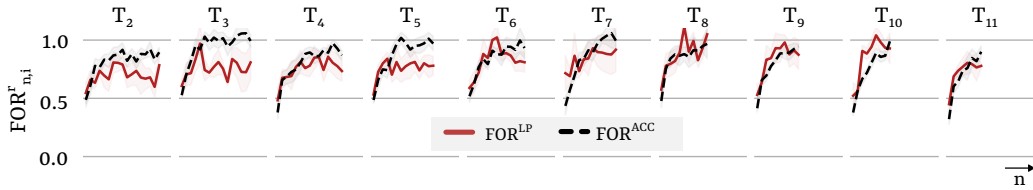

Figure 3: Forgetting at the output level ($\text{FOR}^{\text{ACC}}$) and at the level of representations ($\text{FOR}^{\text{LP}}$), as calculated by Equation 1, on the Mini-ImageNet sequence. When expressed as proportion of the knowledge gained during training on the task, forgetting in the representation is just as catastrophic as forgetting at the output.(Mean $\pm$ standard error)

was learned during training the task. Figure 1 shows why this matters. When not accounting for the initial performance, both examples in this figure forget an equal amount. While when we consider forgetting as the proportion of gained information that was lost, the red example forgets much more. To account for this, we propose to define *relative forgetting* of task $i$ after $n$ new tasks as:

$$\text{FOR}^r_{n,i} = \frac{r_{i,i} - r_{i+n,i}}{r_{i,i} - r_{i-1,i}} \tag{1}$$

Or, in words: relative forgetting is the proportion of knowledge that was gained during training on a task that is then lost after further training on other tasks. When comparing the output accuracy of two continual learning algorithms, our proposed way of measuring forgetting does not often lead to different conclusions, as before training on a task the accuracy is typically low or at chance level. However, for measuring forgetting in representations, our proposal is crucial. To evaluate a representation, some supervised information is always used, hence the initial accuracy will not be zero, but depends on the quality of the representation. While random performance will not change, the quality of the representation can, complicating the analysis further. It is comparable to the difference between the answers to the following two questions before seeing any data: "*Which test samples belong to the unknown category x?*" and "*Given that x looks like this, which other test samples are of category x?*". While the first answer will be random, the second one depends on how good the description, *e.g.* the representation, of *x* is.

In Figure 2, the output accuracy and linear probe accuracy of $T_6$ are shown throughout training. They both peak just after training the task, after which they decrease to a level close to the performance just before the task was trained. Importantly, the two aforementioned differences between output and probing accuracy are apparent in this plot. First, the baseline performance (indicated by $\Delta$) is much higher for the LP measure than for the ACC measure. Secondly, while the output accuracy on $T_6$ does not change by training on the first five tasks, the representation quality does increase. The exact proportions of gained knowledge that are forgotten are hard to compare in this figure, so in

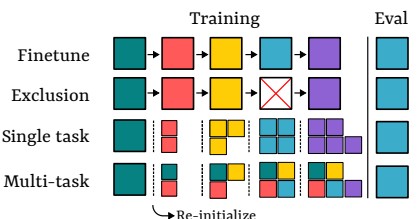

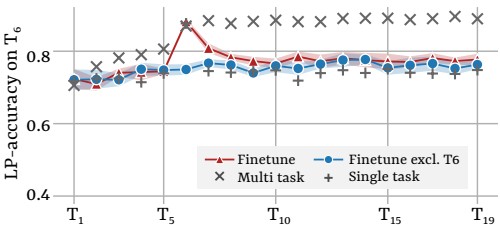

Figure 4: Illustration of the tasks that are trained at each stage for the baselines in Figure 5. Finetune and exclusion continue from the result of the previous task, the others are re-initialized. The white box means no task was trained.

Figure 5: LP-accuracy ($LP_{i,6}$) of a naive finetuning baseline, the exclusion, single and multitask baselines on teh Mini-ImageNet sequence. (Mean $\pm$ standard error).

| Excluded task ($T_e$) | $T_1$ | $T_6$ | $T_{11}$ | $T_{16}$ |
|---|---|---|---|---|
| Finetune | $74.6 \pm 3.2$ | $77.7 \pm 1.6$ | $76.7 \pm 2.5$ | $78.4 \pm 0.9$ |
| Exclusion | $75.2 \pm 2.7$ | $76.3 \pm 1.3$ | $74.8 \pm 2.1$ | $75.3 \pm 1.6$ |
| Single Task | $72.7 \pm 3.7$ | $74.8 \pm 0.8$ | $73.0 \pm 2.9$ | $71.4 \pm 2.5$ |
| Multi Task | $87.3 \pm 2.7$ | $89.0 \pm 1.2$ | $88.6 \pm 1.7$ | $89.4 \pm 1.3$ |

Table 1: Probing accuracy ($LP_{19,e}$ with $e$ the excluded task) at the end of training for tasks mentioned in the columns. Comparing finetuning with exclusion, single task and multi-task training. *e.g.* the second column reports the final values in Figure 5. (Mean $\pm$ standard error)

Figure 3, we show the relative forgetting for both the representation and the output, calculated using Equation 1. For every task $i$, $\text{FOR}_{1,i}$ is as high for the representation as for the output accuracy. For $\text{FOR}_{n,i}$, $n > 1$, it depends on when the task was trained. For early tasks, forgetting of the probe stabilizes, while the output continues to get worse. For the later tasks (see Supplemental for task 12 and more), the representation forgets at least as much as at the output.

Preventing forgetting is one goal of continual learning, forward and positive backward transfer are another: information from one task ideally improves the performance on earlier and later tasks. For representations forgetting and backward transfer can co-occur. Learning from a new task can make a model forget, but at the same time new information can also transfer to an old task. In some sense, this makes the result in Figure 3 only a lower bound to forgetting. It is possible that there is more forgetting, but transfer from other tasks improves the performance at the same time, negating some of the forgetting. Apart from transfer to other tasks, longer optimization with strong augmentations might also cause a model to learn a better representation. To estimate the contribution from transfer from other tasks, we train a model on the same sequence but *without* the evaluated task. This model cannot forget, but it has the transfer from other tasks. Similarly, we train a second model only on the latest task, but increase the number of iterations to match those of the sequential model to evaluate the influence of longer optimization. See Figure 4 for an illustration of their training processes. Figure 5 shows the results and Table 1 contains detailed results with more excluded tasks. Both models that were trained on a sequence of tasks outperform the single task baseline, showing that there is some benefit from training on multiple tasks. This is *knowledge accumulation*: small transfers from other tasks make the final representation better. The exclusion and finetune baseline finally reach nearly the same representation quality. The former cannot forget, so their similarity indicates that in the end, it did not matter much whether a task was trained or not, and a lot of information was forgotten.

## 4 FEATURE FORGETTING REDUCES KNOWLEDGE ACCUMULATION

To answer the second question, whether or not representation forgetting is a problem, we want a baseline that learns in the same way as a continually finetuned model, but that has no forgetting. To achieve this, inspired by Vogelstein et al. (2020) and Yan et al. (2021), we design the *ensemble* baseline. This baseline stores a model copy after every task and concatenates the representation of all these models during evaluation, on top of which the probes are trained. The compute required for

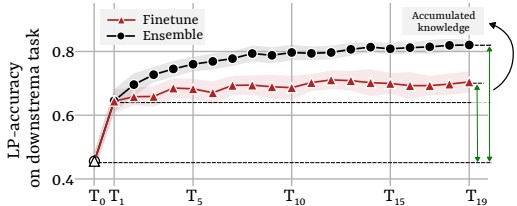 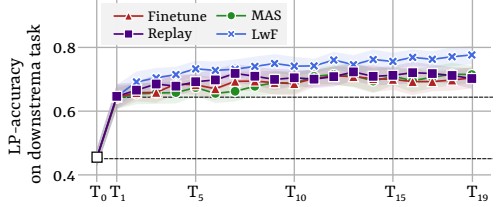

(a) The ensemble baseline accumulates more knowledge than a finetune baseline.

(b) The tested methods of Section 5. LwF has less forgetting, which is confirmed by the better result.

Figure 6: LP-accuracies ($LP_{i,d}$) on a downstream task of Mini-ImageNet. (Mean $\pm$ standard error)

doing inference with this baseline increases linearly with every new task, but it allows us to study the no-forgetting scenario. See Supplemental for more details. Figure 6a shows the probing accuracy on a downstream task. While finetuning accumulates some knowledge, the ensemble baseline clearly accumulates more. We stress again that the finetune and the ensemble baseline learn in the exact same way, the only difference is that the ensemble baseline does not forget. These results thus show that knowledge accumulation is substantially reduced by feature forgetting. A potentially confounding factor in these experiments is that the dimension of the concatenated representation is higher than of the finetuned representation. To control for this, in the Supplemental we use PCA to reduce the dimensionality of the concatenated representation, and show that this does not change our conclusion.

Learning many tasks together resutls in a better general represenation than learning those same tasks sequentially (*e.g.* Zhang et al., 2022; Cha et al., 2022. While it seems that this observation also implies that feature forgetting reduces knowledge accumulation, that conclusion is not actually justified from the observation. It is possible that the representation learned by joint multitask training is better than the representation of finetuning, not because of the absence of forgetting, but because training is done on all tasks at the same time. That is, joint multitask training and finetuning differ not only in terms of forgetting, but also in terms of how they learn.

Continually accumulating knowledge can be a goal on its own, and is often difficult to achieve. Recent works (Janson et al., 2022; Kim & Han, 2023) have shown that recent successful methods for continual learning rely on a pretrained network and remove almost all plasticity. This is a practical solution, but almost entirely depends on the quality of the pretrained representation, without adding new information to the model. Beyond knowledge accumulation, better representations also should result in better continual learners. A better representation can make learning easier as features can be re-used by later tasks and thus do not have to be overwritten (Cha et al., 2021), reducing the risk of additional forgetting. With a few samples (*e.g.* a replay memory), a strong representation can also be used to quickly recover past information, previously referred to as 'fast remembering' (Davari et al., 2022; Hadsell et al., 2020). An important step in enabling knowledge accumulating is thus preventing forgetting of features learned during a task, as shown by the ensemble baseline.

## 5 CAN FEATURE FORGETTING BE PREVENTED?

Over the last years, many methods to alleviate forgetting have been proposed. In this section, we review examples of some of the most important families of methods and evaluate how they deal with feature forgetting and knowledge accumulation. Additionally, we test alternatives to the often used supervised cross-entropy loss in continual learning. The choice of algorithms is not driven by finding the best possible method, but we try to cover the most central ideas, in their simplest form. We test replay with a simple experience replay algorithm with 20 samples per class (ER), parameter regularization using MAS (Aljundi et al., 2018) and functional regularization with LwF (Li & Hoiem, 2017). To test whether our results also hold in different training regimes we also report results using self-supervised learning with BarlowTwins (Barlow) (Zbontar et al., 2021) and contrastive learning with a supervised contrastive loss (SupCon) (Khosla et al., 2020).

Figure 7 shows the forgetting for the tested methods, Table 2 their learning accuracy, or how well they learn new tasks. Replay, MAS and LwF forget at least as much in the output as on their

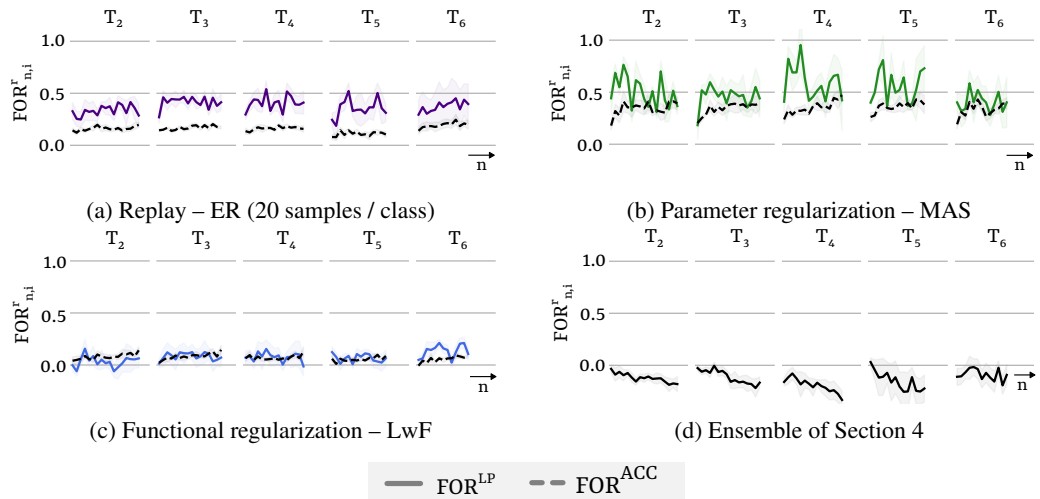

Figure 7: Forgetting as in Equation 1, for $T_2$ to $T_6$ of the tested methods on the Mini-ImageNet sequence. (Mean $\pm$ standard error)

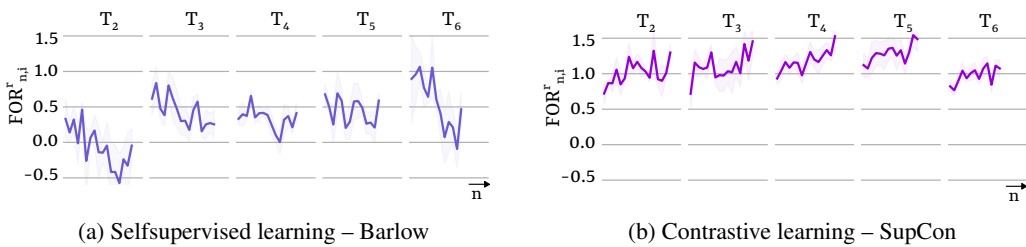

Figure 8: Forgetting for the first 5 tasks using selfsupervised and contrastive losses, instead of the default cross-entropy. (Mean $\pm$ standard error)

representation. LwF prevents a lot of forgetting, both on the representation and at the output level, although it does not close the gap with the ensemble - indicating that it still forgets. Replay and MAS have less forgetting than finetuning, but do not accumulate more knowledge. This is likely the results of their lower learning accuracy, they learn less, so they can accumulate less knowledge. The ensemble shows negative forgetting, because transfer from later tasks further improve performance. Figure 8 reports the other losses. The Barlow Twins baseline has a very 'noisy' forgetting curve. This is likely because the increase in performance during a task is rather small as it does not use any supervised information during training. Surprisingly SupCon forgets even more than it learned. Figure 6b shows the knowledge accumulated for each method, measured by the probing accuracy on a downstream task. Table 2 provides details on the overall improvement of the representations. Similar to the findings in Section 4, representation forgetting prevents knowledge accumulation, which remains true when using continual learning methods. LwF forgets least, and also builds up the most knowledge.

| | Finetune | Concat | Replay | MAS | LwF | Barlow | SupCon |
|---|---|---|---|---|---|---|---|
| $\overline{LP_{i,i}}$ | $86.0 \pm 0.4$ | $86.4 \pm 0.4$ | $81.4 \pm 0.5$ | $79.2 \pm 0.5$ | $85.1 \pm 0.4$ | $78.6 \pm 0.6$ | $85.2 \pm 0.5$ |
| $LP_{1,d}$ | $64.4 \pm 3.8$ | $64.5 \pm 3.7$ | $64.6 \pm 3.9$ | $64.6 \pm 3.6$ | $64.0 \pm 3.3$ | $65.6 \pm 3.5$ | $65.9 \pm 3.7$ |
| $LP_{19,d}$ | $70.4 \pm 3.3$ | $82.0 \pm 1.7$ | $70.2 \pm 3.2$ | $77.6 \pm 2.9$ | $71.5 \pm 2.7$ | $73.9 \pm 1.8$ | $64.1 \pm 3.4$ |
| $LP_{19,d} - LP_{1,d}$ | $6.0 \pm 2.4$ | $17.5 \pm 2.7$ | $5.6 \pm 2.2$ | $7.4 \pm 1.2$ | $13.0 \pm 3.1$ | $8.3 \pm 1.8$ | $-1.8 \pm 1.8$ |

Table 2: Results of the representations of the different methods tested. $\overline{LP_{i,i}}$ denotes the average learning accuracy, *i.e.* the LP-accuracy of a task just after it was trained. The difference between final and initial LP-accuracy measures how much knowledge was accumulated during training. (Mean $\pm$ standard error)

# 6 DISCUSSION

**Representations forget catastrophically**. The results in Section 3 provide compelling evidence that when continually training a model, the information that was learned during a task is catastrophically forgotten. Moreover, in section 5 we find that for the various types of continual learning methods, the representation forgets as much as the observed performance. This seems contradictory to the claims from Davari et al. (2022), but they directly compared output and representation forgetting, not taking the baseline performance and knowledge accumulation into account.

**Forgetting and representation learning are part of the same problem**. In Section 4, we show that a model that is not subject to forgetting, yet learns in the same way as continual finetuning, has the best representation for unseen tasks, with all the discussed benefits. This is again confirmed in Section 5 where especially functional regularization (*e.g.* LwF) has a lot less forgetting, which is reflected by its stronger representation for downstream tasks (see Table 2). For other methods the representations do not significantly improve, although they have lower forgetting. This can be explained by their reduced learning capacity, *i.e.* the accuracy of a newly learned task is less high. This means less is learned, so with the same amount of forgetting there is less knowledge accumulation. See Supplemental for a further details on the learning capacity.

**Role of data and tasks**. As with every machine learning problem, also in this paper there might be a strong dependency on the used data. We tried to reduce this by evaluating all our findings on two datasets (Mini-Imagenet in the main paper, Cifar100 in Supplemental). When comparing performance during training, any metric is always measured on the *same* subset of classes, regardless of the training stage. Some recent works compare results on increasing large sets of classes, which confounds the comparison, as more classes makes for a more difficult problem Cha et al. (2022). Of course, this does not cover all cases. Most importantly, both datasets we used consist of natural images and the trained tasks belong to the same dataset. This makes the opportunity for knowledge transfer arguably larger than when using completely different datasets (not necessarily restricted to natural images). We leave this study for future work, yet hypothesize that with less knowledge accumulation, there is likely even more forgetting, as discussed in Section 3.

**Knowledge accumulation and feature forgetting**. In Section 3 we alluded on the difference between early and later tasks, and how the early tasks seemingly forget less. In Figure 6a we show how the finetune baseline accumulates knowledge and improves on a downstream task. Knowledge accumulation is stronger during the earlier tasks, although it is not a stark difference. Yet it might explain why earlier tasks forget less according to our measure: it is compensated by more knowledge accumulation.

**Future work**. We identify evaluating the current state of the art methods in light of our findings as future work, as well as an analysis on benchmarks that have less related tasks. Combining the benefits of functional regularization with strategies to remove biases in the head can be further investigated to combine the best of both worlds. Finally, as others have reported before, self-supervised and contrastive losses are a promising direction for continual learning (Cha et al., 2021; Davari et al., 2022), yet we showed that these approaches also suffer from feature forgetting.

# 7 RELATED WORK

**Representation learning**. Data rarely come in a format that is adapted to the task we want to perform (Bengio et al., 2013). Except for very simple problems, it is near impossible to directly classify images in their raw pixel representation. For example, many changes in the pixels (*e.g.* translation, rotation, illumination) do not alter the semantics of the image so they should not change the representation. For a long time, researchers have been searching for a representation of images that makes it convenient to solve semantic tasks. Handcrafting features was the standard, *e.g.* (Csurka et al., 2004), but this requires expert knowledge engineering and may not result in optimal features. Since the rise of deep learning, features are more commonly learned by neural networks, directly from the raw data. Both Bengio et al. (2013) and Goodfellow et al. (2016) define *good* representations as ones that make it easier to solve tasks of interest, a definition we adopt. They see deep neural networks as inevitable representation learners, even when this is not explicitly the goal. Neural networks trained to predict image-label pairs indirectly learn a representation where semantically different images are linearly separable in the output of the penultimate layer. Yet representations can

also be learned directly, which can improve robustness, boost generalization, or reduce the need for labeled data (Jing & Tian, 2020).

**Head vs. representation**. The paper proposing iCaRL (Rebuffi et al., 2017) is one of the first continual learning works to explicitly disentangle the representation and head. The head of a model can be relatively well learned with small subsets of data, *e.g.* in the case of classification as a linear layer or with non-parametric approaches like k-nearest neighbors (Wang et al., 2020; Taunk et al., 2019). On the other hand, heads do not transfer well, but quickly become disconnected from the representation when the representation changes while the head is static (Caccia et al., 2021). In the context of continual learning this property has been identified to impact performance severely, and methods updating the last layer only on small memories with balanced data, have shown successes in overcoming much of the observed forgetting (Wu et al., 2019; Zhao et al., 2020).

Recently some continual learning methods explicitly try to foster transfer of knowledge by taking inspiration from advances in representation learning (Jing & Tian, 2020). Some approaches apply contrastive losses (Cha et al., 2021; Mai et al., 2021) and self supervised learning (Marsocci & Scardapane, 2022; Hu et al., 2022; Fini et al., 2022; Rao et al., 2019) to improve continual learning performance, other works take ideas from meta-learning (Javed & White, 2019; Caccia et al., 2020) to learn representations that can easily adapt to new tasks. Lastly, Pham et al. (2021) take inspiration from neuroscience and combine fast and slow learners, *i.e.* supervised and self-supervised modules, in one system.

**Evaluating representation quality**. Effectively leveraging generalization and transfer properties of deep representations is one thing, evaluating their quality is another. As pointed out above, measuring forgetting at the output (the head) of a neural network does not tell us everything about the internal state of a network. Studies that retrain the last layer (Xiong et al., 2019), or a set of deeper layers (Murata et al., 2020), with the earlier layers frozen, hint that representations of lower layers are still useful for seemingly forgotten tasks. However, rather than these layers remembering something specific to the observed tasks, other works interpret this as better generalizability of the lower layers (Ramasesh et al., 2021; Yosinski et al., 2014; Zeiler & Fergus, 2014). Early layers may not seem to forget as much, because their representations are so general that they are almost fully reusable for future tasks, while deeper layers successively encode information more specific to the observed data, that is prone to being overwritten by information of new task's data (Ramasesh et al., 2021).

Davari et al. (2022) and Kim & Han (2023) use linear probes to measure forgetting of the representation in the penultimate layer. Davari et al. (2022) conclude that forgetting is less catastrophic and contrary to (Ramasesh et al., 2021) find that no task-critical information is lost. In contrast Kim & Han (2023) attest severe forgetting in the representation for the set of mechanisms evaluated in both works. The most notable difference in their experimentation setup is that Kim & Han (2023) pre-trainined the model's representation on half the respective dataset in advance. Additionally, the model's ability to incorporate new knowledge (plasticity) is investigated, with the result that most recent continual learning approaches that prevent forgetting at the same time diminish plasticity as well. Zhang et al. (2022) test the performance of a downstream task, showing that finetuning accumulates some knowledge.

## 8 CONCLUSION

In this work we studied how deep neural networks learn and forget representations when continually trained on a sequence of image classification tasks. If forgetting is calculated as the proportion of newly learned knowledge that is forgotten, representations forget at least as much as 'at the output'. Forgetting these representations reduces how much knowledge a model accumulates, as exemplified by the *ensemble* baseline. We further showed that feature forgetting is also observed when training using self-supervised and contrastive losses. Finally, we compared the feature forgetting and knowledge accumulation of different types of continual learning methods, whereby we found that functional regularization can prevent a large portion of representation forgetting. We hope that with the work we present here, future continual learning solutions will be evaluated not only on their output performance, but also on their representation quality and how they prevent feature forgetting.

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
