SUPPLEMENTARY MATERIAL

The supplement material contains additional information on the implementation details and extra results for the experimentation in the main text.

## A  EXPERIMENTATION DETAILS

This section details the training and evaluation of all experiments in the main paper and supplemental material, unless explicitly stated to deviate.

**Data**  MiniImageNet consits of $50,000$ train and $10,000$ test RGB-images of resolution $84 \times 84$ equally divided over $100$ classes. We split this dataset into 20 disjoint tasks such that each task contains five classes. The second benchmark is Split CIFAR-100, which is based on the CIFAR-100 dataset (Krizhevsky et al., 2009) with the same amount of RGB-images and classes as MiniImageNet, but with reduced resolution of $32 \times 32$. We split this dataset into ten disjoint tasks with ten classes each. All experiments are run with five different seeds that also shuffle the class splits over the tasks. See Table 3 and Table 4 for the exact sequences.

**Architecture and optimization**  Throughout this work ResNet-18 (He et al., 2016) is the base architecture for all models. For MiniImageNet we adopt the implementation as default in the pytorch-torchvision (Paszke et al., 2019) library. For CIFAR-100 we employed the slim version of the model as proposed by Lopez-Paz & Ranzato (2017). All networks are trained from scratch, and pre-trained networks are considered future work. The optimization schedules are adjusted with respect to the training criterion. For supervised training with the cross-entropy loss we use an AdamW (Loshchilov & Hutter, 2017) optimizer with static learning rate of $0.001$, weight decay $0.0005$, and beta-values $0.9$ and $0.999$. Each task is trained for 50 epochs with mini-batches of size $128$.

For the SupCon (Khosla et al., 2020) and BarlowTwins (Zbontar et al., 2021) optimization criteria, we stuck to optimization schedules proposed in literature for their application to continual learning. In line with observations by Cha et al. (2021), the SupCon training regime uses an SGD optimizer with momentum $0.9$. The learning rate is scheduled in the same way for every task warming up from $0.0005$ to $0.1$ in the first ten epochs, then annealing by a cosine schedule back to its starting value. The first task is trained for $500$ epochs, all subsequent tasks for $100$ epochs, with a batch size of $256$. The projection network necessary for this objective consists of an MLP with (single) hidden dimension of $512$, projecting to a $128$ dimensional space. Barlow-Twins optimization is aligned to (Marsocci & Scardapane, 2022; Fini et al., 2022). We use an Adam optimizer (Kingma & Ba, 2015) with learning rate $0.0001$ and weight decay $0.0005$. We train $500$ epochs for each task with batch size of $256$. Again, the projection head is an MLP but with two hidden layers, and hidden and final projection dimension of $2048$. All methods use the same augmentations, see below.

**Probe optimization**  To quantify the quality of the representation we apply probes based on linear- and k-nearest neighbors- ($k$NN) classifiers. Linear classifiers consist of a single linear layer. In the optimal probing case, reported mostly throughout the work, it is optimized with access to all training data. Linear probes are optimized analog to Cha et al. (2021). Keeping a batch-size of $128$, we use SGD with momentum of $0.9$ and no weight decay for $100$ epochs. The learning rate of $0.1$ is decaying at epochs 60, 75, and 90 by a factor of $0.2$. Similarly, $k$NN uses all training data to evaluate the representations.

**Continual learning mechanisms**  LwF and MAS are using a value of $\lambda = 1.0$ as advocated by its original authors. Replay uses a random selection of 20 exemplars per class. The weight of the loss on replayed samples is increased proportionally to the number of previously observed tasks, to prevent favoring the current task in the optimization. An upper bound is reported by jointly training the model on all observed data. For our lower-bound we want to document the impact the singled out tasks have. This we achieve by re-initializing the model before training a new task, but allowing the new task to train for as many iterations as a continual model would have, *e.g.* 50 epochs for the first task, then 100 for the second, and so on. By design this model has zero transfer of knowledge, and we will refer to it as 'Single task' baseline.

**Augmentations**  In all experiments we use the data augmentation pipeline from SimCLR Chen et al. (2020). The augmentations pipeline consists of random crops and horizontal flips, color-jitter (brightness=0.4, contrast=0.4, saturation=0.2, hue=0.1), random grayscaling (p=20%) and Gaussian blur using a kernel of size 9 and sigma range $0.1$ to $0.2$. In PyTorch, the augmentations are defined as follows:

```
from torchvision.transforms import *

RandomHorizontalFlip(p=0.5),
RandomResizedCrop(size=(32, 32), scale=(0.2, 1.0)),
RandomApply(
    [ColorJitter(brightness=0.4, contrast=0.4, saturation=0.2, hue=0.1)], p=0.8),
RandomGrayscale(p=0.2),
RandomApply([
    GaussianBlur(kernel_size=input_size[0]//20*2+1, sigma=(0.1, 2.0))], p=0.5)
```

## B  RELATIVE FORGETTING: EXTRA RESULTS

Figure 3 only shows the relative forgetting in the Mini-ImageNet task sequence. For completness, we report here the relative forgetting of all tasks.

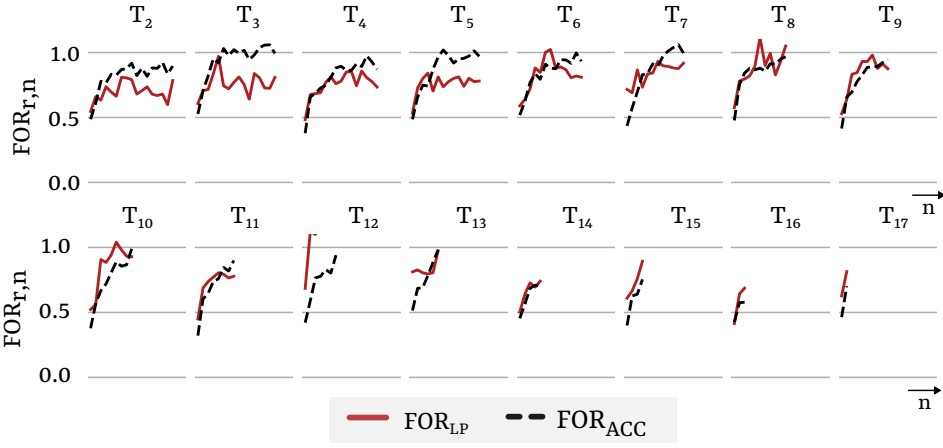

Figure 9: Representation and observed forgetting using linear probes for all tasks in Mini-ImageNet using finetuning (except first and last, for which we cannot calculate relative forgetting)

## C  ENSEMBLE: FURTHER DETAILS

The ensemble method trains stores a model copy every after every task. Each of these models output a representation $f_i$ with dimension $k$ for the input data. During training, only the model of the task is used and the others ae frozen. Before evaluating and training of the linear probes during evaluation, all of the representations $f_t$ are concatenated to form one large representation $f = [f_1, f_2 \cdots f_t]$. On top of this large representation a linear layer with input dimension $tk$ is trained, instead of just $k$ for the finetuned model.

To mitigate the influence of the higher dimension, for which it might be easier to find linearly separable features, we add a dimension reduction to lower the dimension back to $k$. We do this by projecting the features of the ensemble on the top-$k$ most significant PCA dimensions. The results are shown in Figure 10. The reduced ensemble performs a bit worse than the full ensemble, yet still significantly better than finetuning.

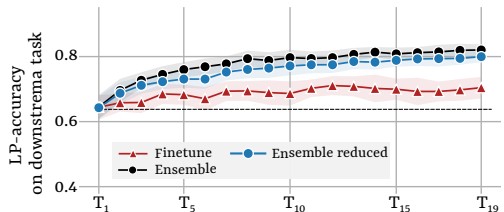

Figure 10: LP-accuracies of finetuning, the ensemble baseline and its reduced version, as explain in Section C

# D  RESULTS ON CIFAR100

To reduce the dependency on only having experiments on a single dataset, we report our main results here also on CIFAR100. The results on CIFAR100 follow the same general trends as those on Mini-ImageNet in the main paper. The largest difference is that the effects are sometimes smaller, due to the shorter task sequence.

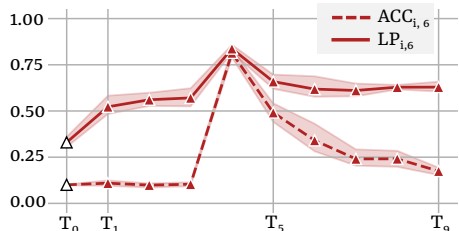

Figure 11: Linear probe and output accuracy of $T_3$ during the entire CIFAR100 sequence.

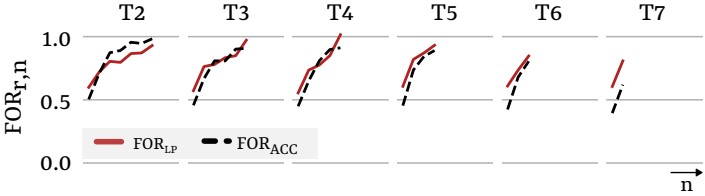

Figure 12: Representation and observed forgetting using linear probes for all tasks in CIFAR100 using finetuning (except first and last task, for which we cannot calculate relative forgetting)

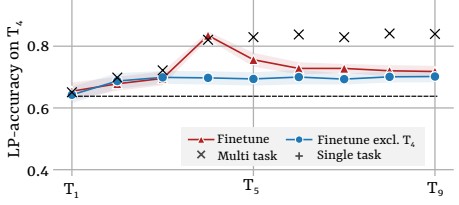

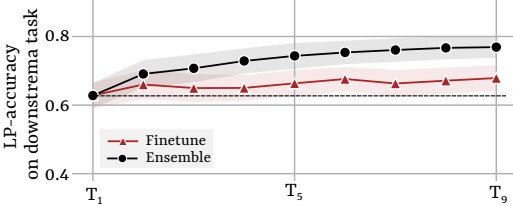

Figure 13: Finetune, exclusion, single task and multi task with CIFAR100.

Figure 14: Comparing the ensemble and finetuning on CIFAR100.

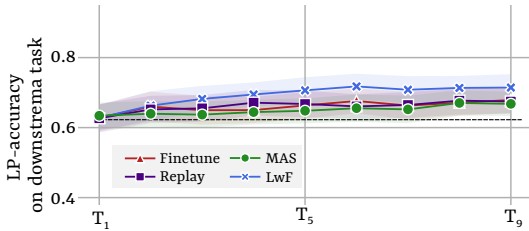

Figure 15: LP-accuracies on a downstream task of CIFAR100.

# E EVALUATION WITH $k$NN

In this section we report the most important results of the main paper using $k$NN instead of using linear probes. This has the benefit that it there are no hyperparameters to tune and does not depend on the optimization used. We report it here for completness, and keep the linear probes in the main paper as this is how preivous papers reported their results Davari et al. (2022); Cha et al. (2022); Zhang et al. (2022). In general, the results in Figure 16, 17, 18 and 19 follow the same trends as observed in the main paper, with the main difference that the absolute values are lower, likely due to the suboptimalitiy of $k$NN compared to linear probe optimization.

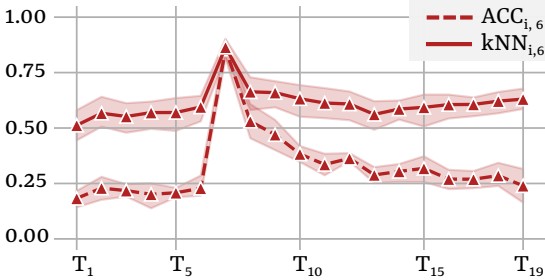

Figure 16: $k$NN and output accuracy of $T_6$ during the entire Mini-ImageNet sequence.

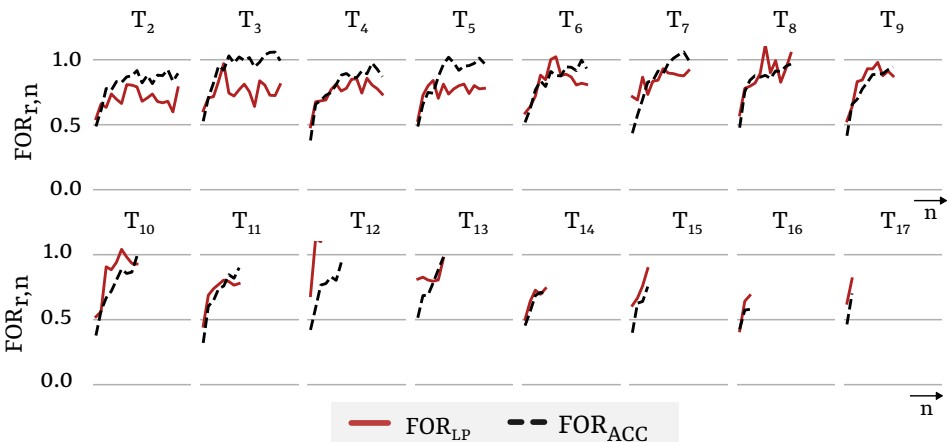

Figure 17: Representation and observed forgetting using $k$NN for all tasks in Mini-ImageNet (except last and first, for which we cannot calculate relative forgetting)

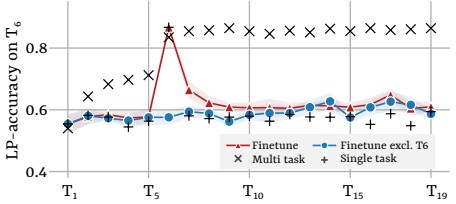 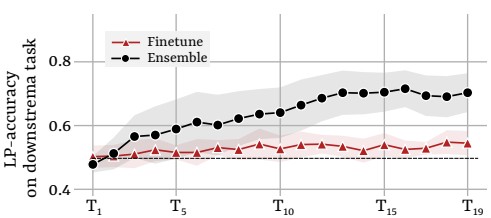

Figure 18: Finetune, exclusion, single task and multi task using *k*NN.

Figure 19: Comparing the ensemble and finetuning using *k*NN.

## F    DETAILED TASK SEQUENCE INFORMATION

In Table 3 and Table 4 we report the exact task sequences used in the experiments in the main paper. These are chosen at random, but consistent in all experiments. The randomness of the tasks also means that there difficult varies quite a bit, which explais some of the higher standard errors in the experiments reported.

| idx | Synset | Synset name | idx | Synset | Synset name | idx | Synset | Synset name |
|---|---|---|---|---|---|---|---|---|
| 0 | n01532829 | house_finch | 33 | n03400231 | frying_pan | 66 | n02981792 | catamaran |
| 1 | n01558993 | robin | 34 | n03476684 | hair_slide | 67 | n03980874 | poncho |
| 2 | n01704323 | triceratops | 35 | n03527444 | holster | 68 | n03770439 | miniskirt |
| 3 | n01749939 | green_mamba | 36 | n03676483 | lipstick | 69 | n02091244 | Ibizan_hound |
| 4 | n01770081 | harvestman | 37 | n03838899 | oboe | 70 | n02114548 | white_wolf |
| 5 | n01843383 | toucan | 38 | n03854065 | organ | 71 | n02174001 | rhinoceros_beetle |
| 6 | n01910747 | jellyfish | 39 | n03888605 | parallel_bars | 72 | n03417042 | garbage_truck |
| 7 | n02074367 | dugong | 40 | n03908618 | pencil_box | 73 | n02971356 | carton |
| 8 | n02089867 | Walker_hound | 41 | n03924679 | photocopier | 74 | n03584254 | iPod |
| 9 | n02091831 | Saluki | 42 | n03998194 | prayer_rug | 75 | n02138441 | meerkat |
| 10 | n02101006 | Gordon_setter | 43 | n04067472 | reel | 76 | n03773504 | missile |
| 11 | n02105505 | komondor | 44 | n04243546 | slot | 77 | n02950826 | cannon |
| 12 | n02108089 | boxer | 45 | n04251144 | snorkel | 78 | n01855672 | goose |
| 13 | n02108551 | Tibetan_mastiff | 46 | n04258138 | solar_dish | 79 | n09256479 | coral_reef |
| 14 | n02108915 | French_bulldog | 47 | n04275548 | spider_web | 80 | n02110341 | dalmatian |
| 15 | n02111277 | Newfoundland | 48 | n04296562 | stage | 81 | n01930112 | nematode |
| 16 | n02113712 | miniature_poodle | 49 | n04389033 | tank | 82 | n02219486 | ant |
| 17 | n02120079 | Arctic_fox | 50 | n04435653 | tile_roof | 83 | n02443484 | black-footed_ferret |
| 18 | n02165456 | ladybug | 51 | n04443257 | tobacco_shop | 84 | n01981276 | king_crab |
| 19 | n02457408 | three-toed_sloth | 52 | n04509417 | unicycle | 85 | n02129165 | lion |
| 20 | n02606052 | rock_beauty | 53 | n04515003 | upright | 86 | n04522168 | vase |
| 21 | n02687172 | aircraft_carrier | 54 | n04596742 | wok | 87 | n02099601 | golden_retriever |
| 22 | n02747177 | ashcan | 55 | n04604644 | worm_fence | 88 | n03775546 | mixing_bowl |
| 23 | n02795169 | barrel | 56 | n04612504 | yawl | 89 | n02110063 | malamute |
| 24 | n02823428 | beer_bottle | 57 | n06794110 | street_sign | 90 | n02116738 | African_hunting_dog |
| 25 | n02966193 | carousel | 58 | n07584110 | consomme | 91 | n03146219 | cuirass |
| 26 | n03017168 | chime | 59 | n07697537 | hotdog | 92 | n02871525 | bookshop |
| 27 | n03047690 | clog | 60 | n07747607 | orange | 93 | n03127925 | crate |
| 28 | n03062245 | cocktail_shaker | 61 | n09246464 | cliff | 94 | n03544143 | hourglass |
| 29 | n03207743 | dishrag | 62 | n13054560 | bolete | 95 | n03272010 | electric_guitar |
| 30 | n03220513 | dome | 63 | n13133613 | ear | 96 | n07613480 | trifle |
| 31 | n03337140 | file | 64 | n03535780 | horizontal_bar | 97 | n04146614 | school_bus |
| 32 | n03347037 | fire_screen | 65 | n03075370 | combination_lock | 98 | n04418357 | theater_curtain |

Table 3: The classes included in Split MiniImagenet, with their index, (which is not general, but used in the task splits), their synsets and their name.

|  | Seed 42 | Seed 52 | Seed 62 | Seed 72 | Seed 82 |
|---|---|---|---|---|---|
| T1 | 83 - 53 - 70 - 45 - 44 | 82 - 8 - 44 - 19 - 2 | 76 - 48 - 62 - 80 - 29 | 76 - 82 - 43 - 16 - 84 | 72 - 33 - 58 - 2 - 55 |
| T2 | 39 - 22 - 80 - 10 - 0 | 73 - 37 - 89 - 67 - 18 | 99 - 60 - 89 - 39 - 69 | 95 - 78 - 91 - 30 - 22 | 84 - 54 - 75 - 28 - 40 |
| T3 | 18 - 30 - 73 - 33 - 90 | 4 - 92 - 83 - 24 - 14 | 14 - 74 - 59 - 87 - 55 | 1 - 96 - 25 - 81 - 62 | 39 - 15 - 41 - 12 - 35 |
| T4 | 4 - 76 - 77 - 12 - 31 | 93 - 90 - 84 - 81 - 66 | 40 - 46 - 54 - 92 - 7 | 5 - 18 - 63 - 14 - 24 | 23 - 49 - 91 - 32 - 38 |
| T5 | 55 - 88 - 26 - 42 - 69 | 40 - 72 - 56 - 36 - 51 | 6 - 32 - 77 - 27 - 63 | 23 - 75 - 9 - 60 - 27 | 64 - 68 - 6 - 92 - 18 |
| T6 | 15 - 40 - 96 - 9 - 72 | 50 - 68 - 88 - 55 - 57 | 96 - 33 - 49 - 25 - 68 | 83 - 20 - 90 - 55 - 36 | 48 - 47 - 13 - 89 - 79 |
| T7 | 11 - 47 - 85 - 28 - 93 | 27 - 29 - 80 - 3 - 94 | 26 - 94 - 38 - 85 - 98 | 4 - 10 - 77 - 93 - 33 | 96 - 22 - 34 - 81 - 63 |
| T8 | 5 - 66 - 65 - 35 - 16 | 53 - 62 - 87 - 52 - 95 | 61 - 43 - 93 - 15 - 28 | 58 - 35 - 97 - 11 - 59 | 53 - 85 - 14 - 50 - 44 |
| T9 | 49 - 34 - 7 - 95 - 27 | 70 - 12 - 1 - 97 - 48 | 36 - 2 - 42 - 75 - 31 | 56 - 98 - 47 - 86 - 38 | 24 - 61 - 11 - 0 - 21 |
| T10 | 19 - 81 - 25 - 62 - 13 | 60 - 47 - 65 - 10 - 41 | 22 - 56 - 3 - 67 - 19 | 85 - 66 - 49 - 41 - 87 | 10 - 59 - 90 - 71 - 56 |
| T11 | 24 - 3 - 17 - 38 - 8 | 17 - 96 - 9 - 49 - 30 | 20 - 90 - 50 - 84 - 66 | 42 - 99 - 57 - 0 - 6 | 17 - 76 - 1 - 95 - 70 |
| T12 | 78 - 6 - 64 - 36 - 89 | 38 - 58 - 0 - 26 - 21 | 70 - 97 - 4 - 64 - 44 | 70 - 13 - 50 - 40 - 68 | 94 - 37 - 5 - 4 - 26 |
| T13 | 56 - 99 - 54 - 43 - 50 | 31 - 15 - 75 - 25 - 6 | 82 - 47 - 95 - 41 - 51 | 48 - 73 - 37 - 8 - 39 | 60 - 20 - 45 - 98 - 74 |
| T14 | 67 - 46 - 68 - 61 - 97 | 74 - 59 - 64 - 43 - 34 | 23 - 5 - 79 - 88 - 34 | 32 - 3 - 89 - 51 - 44 | 62 - 57 - 73 - 97 - 87 |
| T15 | 79 - 41 - 58 - 48 - 98 | 20 - 77 - 7 - 78 - 71 | 16 - 35 - 52 - 71 - 72 | 17 - 54 - 15 - 67 - 2 | 46 - 51 - 7 - 82 - 83 |
| T16 | 57 - 75 - 32 - 94 - 59 | 22 - 39 - 63 - 76 - 85 | 57 - 12 - 1 - 13 - 86 | 31 - 52 - 61 - 34 - 71 | 19 - 88 - 9 - 8 - 52 |
| T17 | 63 - 84 - 37 - 29 - 1 | 79 - 45 - 61 - 42 - 46 | 78 - 8 - 21 - 91 - 83 | 64 - 92 - 65 - 53 - 28 | 30 - 65 - 16 - 36 - 69 |
| T18 | 52 - 21 - 2 - 23 - 87 | 54 - 91 - 16 - 5 - 33 | 10 - 0 - 65 - 73 - 37 | 72 - 80 - 12 - 45 - 21 | 25 - 67 - 43 - 29 - 42 |
| T19 | 91 - 74 - 86 - 82 - 20 | 35 - 98 - 69 - 32 - 99 | 45 - 30 - 17 - 53 - 58 | 29 - 7 - 26 - 79 - 69 | 78 - 80 - 31 - 86 - 93 |
| Downstream task | 60 - 71 - 14 - 92 - 51 | 86 - 23 - 13 - 11 - 28 | 11 - 9 - 81 - 24 - 18 | 94 - 74 - 46 - 19 - 88 | 77 - 27 - 99 - 66 - 3 |

Table 4: Task splits used in the results with Split MiniImagenet. The indices correspond to the classes listed in Table 3. Results reported on Split MiniImagenet average over these 5, randomly determined, task sequences.