# OpenReview forum: "Knowledge Accumulation in Continually Learned Representations and the Issue of Feature Forgetting"
_ICLR.cc/2024/Conference — ICLR 2024 Conference Withdrawn Submission_

### Official Review · Reviewer_BtF4 · 2023-10-29

**Soundness:** 2 fair
**Presentation:** 2 fair
**Contribution:** 1 poor
**Rating:** 3
**Confidence:** 4

**Summary:**

This paper analyzes the catastrophic forgetting issue in continual learning.
The authors focus on forgetting in the *representation*, which is narrowly defined as the last activation right before the final output.
The main findings are summarized as follows.
1. Catastrophic forgetting does occur in the representation.
2. This forgetting harms the final performance.

**Strengths:**

The text is easy to follow.
The main findings are clearly stated at the beginning.

**Weaknesses:**

### Novelty

The major weakness of this work is that the main findings are not new.
I think the two main findings are the most basic assumptions of continual learning.
Personally, the most surprising part of this paper was that there are several works claiming that forgetting is minimal in the representation.
But if forgetting in the representation is negligible, why would the entire field of continual learning exist?


### Limited Scope of Analysis

The analyses in this work were conducted in quite narrow settings.
The authors focused exclusively on the last activation of a network for classification tasks.
Furthermore, they concentrated solely on the offline continual learning scenarios, excluding online continual learning from their scope.

### Writing

While I didn't have much trouble understanding the paper, there is room for improvement in the overall writing, particularly in terms of grammar.

**Questions:**

N/A

---

> ### Author Response · Authors · 2023-11-20
>
> We thank the reviewer for raising a very important point, that is **“the two main findings are the most basic assumptions of continual learning”** and take it as a chance to also address: **“The major weakness of this work is that the main findings are not new. “**, re-emphasizing the novelty of our work.
>
> While we agree with the reviewer’s forwarded assumptions on how forgetting and learning should interact in the continually learning model, we would like to point out that these assumptions were, to the best of our knowledge, never directly demonstrated to hold.
> In this work we address the claim, which has repeatedly been made in the recent literature, that forgetting in representations is "not catastrophic". An innate robustness of representation or representation learning methods to forgetting would be a promising direction for continual learning research. Earlier work in the field might not have sufficiently observed this robustness because it was mostly concerned with investigating forgetting through the lens of measuring final (average) performance of the continual learner. In contrast, recent works with explicit focus on the accumulated representation in continual learning partially do find such a robustness. The contribution of our work is to clarify and settle this disagreement. We do so with a novel perspective and detailed analysis that first shows indisputably that representations do forget catastrophically and secondly illustrate the direct connection to impaired continual knowledge accumulation. Despite reasonable but intuitive arguments derived from earlier works, to the best of our knowledge, such an analysis was yet missing from literature.
>
> **“The authors focused exclusively on the last activation of a network for classification tasks”.**
> It is true that in this work we did focus on the representation resulting from the last activation of the network. And more fine grained investigation of the representations at different depths is desirable future work. However, for the points to make in this work we felt it to be out of scope. We hope that future empirical analysis will successively reveal these traits.
>
> **“Solely on offline continual learning scenarios.”**
> We stand by our analysis repeating the same argument as for the previous point. Certainly, it can be quite interesting to similarly consider an online perspective, however, to prove the points of our work we deemed the upper-bound analysis of sequential offline training sufficient.

---

> > ### Comment · Reviewer_BtF4 · 2023-11-22
> >
> > Thank you for the response.
> >
> > Unfortunately, I still think this work does not contribute much to the CL research community.
> > I'll be sticking with my initial assessment.

---

### Official Review · Reviewer_gcjs · 2023-10-31

**Soundness:** 3 good
**Presentation:** 2 fair
**Contribution:** 3 good
**Rating:** 6
**Confidence:** 5

**Summary:**

This paper tries to explore whether neural networks suffer from catastrophic forgetting at the level of representations. This paper focuses on two questions: Do continually trained representations forget catastrophically, and Does it matter that these representations are forgotten. To answer these questions, the main contributions of this paper are summarized as"

a. This paper shows that continually learned respresentations do forget catastrophically.

b. The respresentation forgetting negatively affects knowledge accumulation.

c. This paper also consider feature forgetting and knowledge accumulation in continual learning methods.

d. This paper explores the feature forgetting with self-supervised and contrastive losses.

**Strengths:**

a. This work is novel and explores whether neural networks suffer from catastrophic forgetting at the level of representations.

b. This paper is well-written and easy to follow.

c. Extensive experiments. I appreciate that this paper provides extensive experiments to show the effectiveness of the proposed method.

**Weaknesses:**

Although the experimental phenomenon presented in this paper is very interesting, it is essentially an experimental work. It is hard to determine whether the conclusions drawn in this paper are widespread or only based on the experimental settings (models, datasets) used in this work. However, this type of work is OK and interesting.

**Questions:**

In Section 3, this paper argues that before learning task $t$, the model contains different level of information of task $t$ in different scenarios (red and blue lines). It is noteworthy that all tasks in CL are disjoint (mentioned in Section 2) and the model starts from scratch (mentioned in Appendix A). How does the model achieve the knowledge of task $t$ before learning task $t$ and why are you confirmed that this knowledge achievement is valid? Please discuss more about it and it is better to provide theoretical and experimental support.

---

> ### Author Response · Authors · 2023-11-20
>
> We thank the reviewer for sharing our interest in this research direction and appreciating the novelty we provide.
>
> **“How does the model achieve the knowledge of task t before learning task t and why are you confirmed that this knowledge achievement is valid?”.**
> To answer briefly: In our selected settings we always observe natural images which share quite a large portion of their distribution which can nicely transfer. However, we agree with the intuition behind the question and likewise hypothesize that in other scenarios, where data is not sampled from overlapping distributions, (absolute) forgetting might be much more severe.

---

### Official Review · Reviewer_acbx · 2023-11-02

**Soundness:** 2 fair
**Presentation:** 1 poor
**Contribution:** 1 poor
**Rating:** 3
**Confidence:** 5

**Summary:**

The paper investigates the relationship between representation feature forgetting and knowledge accumulation during continual learning. The paper suggests relative forgetting that is very similar to Backward Transfer, yet dividing it with the performance improvement obtained by training the target task. To demonstrate the authors' claims, the paper provides multiple empirical results. In summary, they find that the continual learning models consistently forget the features severely, and this interferes with knowledge accumulation when learning new tasks. Additionally, this feature forgetting can be alleviated through adopting various continual learning approaches.

**Strengths:**

The authors scrab various claims on catastrophic forgetting during continual learning from multiple literatures. And suggests a new relative forgetting metric.

**Weaknesses:**

- Regarding feature forgetting, the paper simply repeats the observations of prior/conventional literature on continual learning: forgetting occurs, and it matters the performance of the model. In that sense, the suggested relative forgetting metric does not show any distinguished observations on existing metrics like Backward Transfer and Averaged Forgetting.

- Limited contribution: Although the paper is dedicated to studying well-known and sufficiently analyzed challenges in continual learning fields, the evaluation tasks, domains, models, method types (e.g., rehearsal-/architecture-/regularization-/prompt-based approaches), ..., are limited, and it is hard to catch 'new'/'novel' insights. - Most observations resort to image-based benchmark classification tasks. There are various continual learning approaches in vision/language/multimodal domains with diverse tasks, segmentation/object detection/generation/text classification/(visual) question answering, etc.

- Presentation/writing can be further improved. It seems to include repeated claims and unnecessary sentences. For example, the first paragraph in Section 3 is about what is 'catastrophically', but this paragraph is not aligned with the overall flow and arguments of the paper. the word 'catastrophic' simply indicates critically bad, or severe, and no more implication.

- The faithfulness/benefit of relative forgetting is not clearly described. Regardless of initial performance on target tasks in continual learning, the model contains the most beneficial representations of the task when its performance is the highest during continual learning, and the degenerated performance can be considered as knowledge loss, i.e., forgetting. As shown in the paper, this new metric shows a similar tendency to existing forgetting/backward transfer metrics without new insights, It is not clear why we need to care about the 'relative' forgetting.

- In section 4, the suggested ensemble baseline violates the conventional continual learning setting and is clearly different from the typical continual learning model. Let us store N backbone models by training N past tasks sequentially, the authors concatenate all features on evaluation data from these models and propagate the concatenated features to the classifier. Here, the input dimension of the classifier is different from the base continual learning model (proportional to the number of past tasks (i.e., stored models)), and this means the trainable parameters are N times larger. This is totally different model, and evaluation analyses with the assumption that 'the base continual learning model and ensemble models learn the continual learning tasks in the same way' may not be correct.

- Aligned with the second weakness, I strongly recommend that the authors provide further clear contributions against earlier works that extensively study representational forgetting and transferability in continual learning methods [1,2]. In particular, [2] also observed different behaviors among supervised, self-supervised, and contrastive continual learning in view of representation forgetting and knowledge accumulation.

[1] Chen et al., "Is forgetting less a good inductive bias for forward transfer?" ICLR 2023.
[2] Yoon et al., "Continual Learners are Incremental Model Generalizers", ICML 2023.

**Questions:**

.

---

> ### Author Response · Authors · 2023-11-20
>
> We thank the reviewer for their feedback, and for pointing us to two useful references.
>
> **“The paper simply repeats the observation of prior literature on continual learning: forgetting occurs, and it matters the performance of the model.”**
> Earlier works mostly studied forgetting through the lens of measuring final (average) performance of the continual learner. Our work aligns with a surge of recent works that set focus on exploring learning and forgetting at the representation level. Different from the former works, when regarding representations we reviewed published evidence that forgetting may be less severe than previously assumed. We emphasize that the study conducted in our work analyzes the progression of learning and forgetting with respect to each task’s data individually, especially by also differentiating between preceding and newly learned information in the representation. The latter is done with our relative forgetting metric. We find this nuance is necessary to unravel indisputably that representations do forget catastrophically - and further - that forgetting impairs continual knowledge accumulation. Partial aspects may be expectable in hindsight or hinted at in concurrent work, but to the best of our knowledge we are the first to provide indisputable analysis that illustrates catastrophic dynamics of forgetting in representations and its impact on knowledge accumulation by considering a perspective that was yet missing from literature.
>
> Further, we would also like to answer some of the more detailed forwarded concerns.
> * **“experimentation is limited”:** Our experimentation is aligned to the protocols of the works we compare to, in settings where previously mentioned evidence was first brought up. Of course we hope our proposed analysis is carried over to future investigation settings.
>
> * **“relative forgetting is very similar to backward transfer [...] this new metric shows a similar tendency to existing forgetting/backward transfer metrics without new insights”:** The relative measure of forgetting we propose is, albeit mathematically simple, better representing forgetting with respect to prior learning rather than considering absolute values that can be deceiving. This core motivation and merit is illustrated in Figure 1 and discussed in Section 3.
>
> * **“the suggested ensemble baseline violates the conventional continual learning setting [...] the input dimension of the classifier is different from the base continual learning model [...] the trainable parameters are N times larger. This is totally different model and evaluation analysis with the assumption that 'the base continual learning model and ensemble models learn the continual learning tasks in the same way' may not be correct. ”:**
> We would like to point out that, during training, the ensemble baseline learns **in exactly the same way** as the fine-tuned model. For example, when training on the $n$-th task, the $n$-th submodel of the ensemble baseline (i.e., the submodel being trained on the $n$-th task), is exactly the same as the fine-tuned model when it is being trained on the $n$-th task. The only difference is in evaluation. During evaluation, we do not only use the representation of the latest model, but we also use all the models that we learned along the way. Note that because learning was exactly the same, all these intermediate models do not contain any information that was not at some point also learned by the fine-tuned model. So the ensemble model does not use any additional information; it is just a way to keep track of information that is forgotten by the fine-tuned model. It is true that a possible confounding factor is the dimensionality of the concatenated representation of the ensemble model, but please note that we control for this in the Appendix using dimensionality reduction techniques.

---

### Official Review · Reviewer_1Fn7 · 2023-11-07

**Soundness:** 3 good
**Presentation:** 2 fair
**Contribution:** 2 fair
**Rating:** 3
**Confidence:** 4

**Summary:**

This paper studies the quality of learned representations in continual learning. With the help of two new metrics -- linear probe accuracy and relative forgetting, it is shown that representation learning also suffers from catastrophic forgetting in both continual supervised learning and continual self-supervised learning, and thus reduces the overall task performance.

**Strengths:**

- The two proposed metrics --- linear probe accuracy and relative forgetting --- are useful for the community.
- Experiments are performed in both continual supervised learning and continual self-supervised learning.

**Weaknesses:**

- The writing and presentation can be significantly improved. For example, Figure 4 is quite confusing. What does each square mean? What does each color represent? How about the sizes of the squares?
- Lack of deep analysis. This is my major concern. Since this paper does not propose new methods or new theories, I would expect to see more insights about continual representation learning, which the paper does not provide much. The main conclusions are within expectation and well known in the continual learning community. Knowledge accumulation and feature forgetting are another expression of the stability-plasticity dilemma. I would encourage authors to improve this work by providing deeper analysis. For example,
  - In the abstract, it is mentioned that "Some studies ascribe a certain level of innate robustness to representations, that they only forget minimally and no critical information, while others claim that representations are also severely affected by forgetting." Why does the contradiction exist? With two proposed metrics, can we explain and verify this contradiction effectively and directly?
  - Different methods are tested to prevent representation forgetting. However, there is a lack of analysis or discussion to explain why method A is better than method B.

**Questions:**

See Weaknesses.

---

> ### Author Response · Authors · 2023-11-20
>
> We thank the reviewer for evaluating the proposed metrics as useful for the community and the appreciation of the experimentation.
>
> **“The main conclusions are within expectation and well known”.**
> Most prior works investigated forgetting through the lens of measuring final (average) performance of the continual learner. Recently, a body of literature shifted focus to exploring representations in continual learning, and partially - different to prior findings - raised evidence for an innate robustness to forgetting. We settle this disagreement with a novel perspective and analysis that first indisputably shows that representations do forget catastrophically and secondly illustrates the direct connection to impaired continual knowledge accumulation. To the best of our knowledge such a study was yet missing from literature.
>
> **“Why does the contradiction exist?”**
> We provide one explanation to this question in Figure 1, where we contrast absolute forgetting measures with our relative one. If one reasons with absolute forgetting, it may seem like not everything is forgotten at the representation level. When using relative forgetting instead it becomes clear that there is no contradiction.

---

> > ### Comment · Reviewer_1Fn7 · 2023-11-20
> > **Reply**
> >
> > Thank you for your comments. I read other reviews and decided to maintain my previous score.